# Analysis of Influencing Factors in Purchasing Electric Vehicles Using a Structural Equation Model: Focused on Suwon City

**Sukhee Kim [1], Jungyoon Choi [2], Yongju Yi [2] and Hyungjun Kim [1,*]**

[1] Department of Urban Space Research, Suwon Research Institute, Gweonseon-gu, Suwon 16429, Korea; sukheek@suwon.re.kr
[2] TOD-Based Sustainable Urban/Transportation Research Center, Yeongtong-gu, Suwon 16499, Korea; cjungy@ajou.ac.kr (J.C.); srzr2001@ajou.ac.kr (Y.Y.)
* Correspondence: dukkubi0512@suwon.re.kr; Tel.: +82-31-220-8070

**Abstract:** The global automobile market is promoting the introduction of eco-friendly vehicles such as electric vehicles and hydrogen vehicles. However, disadvantages such as expensive prices and limited mileage compared to internal combustion engine vehicles have become obstacles to the expansion of eco-friendly vehicles. Therefore, in this study, a survey was conducted on the purchase of electric vehicles for citizens of Suwon. Using the survey data, a structural equation model was constructed to analyze the factors affecting the purchase of electric vehicles, which are eco-friendly vehicles. The results indicate that a lack of information and government policy on EV, the level of EV recognition and subsidy policy do not have an effect on EV purchase. However, charging infrastructure, battery performance and safety, operating conditions including ramps or use of heaters and air conditioners, subsidy effects and charging services demonstrate positive effects on EV purchase. Using direct and indirect effect analysis, the study shows that higher government subsidy and visiting charging services are the two most influential factors on EV purchase, followed by EV driving environment, charging infrastructure, battery performance and safety, and a lack of information and electric vehicle supply policy.

**Keywords:** electric vehicle; electric vehicle activation; purchase factors; eco-friendly vehicles; structural equation model

## 1. Introduction

Emissions of greenhouse gases and global climate change are among the biggest issues on earth; environmental pollution caused by emissions in the transportation area has especially been a problem. As a solution to these issues, many countries are implementing policies to transition to zero-emission vehicles such as electric cars and hydrogen fuel cell vehicles. Moreover, automobile industries are also changing the paradigm of manufacturing production from internal combustion engine-based vehicles to other environmentally friendly vehicles that are sustainable to respond to global movements.

South Korea is also participating in the clean energy movement as a corresponding solution to severe air pollution *By* making policies to nationally expand the use of electric cars. The South Korean government offers a tax reduction and government subsidy to its citizens when they purchase an electric car along with other incentives such as offering discounts at municipal parking lots.

A total of 133 electric cars were registered in Suwon city, composed of 20 city government vehicles and 113 privately owned vehicles according to data collected in 2017. This was approximately 0.03% of the total registered cars in Suwon city and 44.3% of the city policymakers' goal for their expansion. Therefore, it is vital to examine the specific factors that would make consumers be willing to buy electric cars. The higher price compared to internal combustion engine cars and the limited driving range per electrical charging session are indicated as major obstacles to purchasing electric cars for consumers.

Recent studies have been focused on choice models to estimate the demand for electric vehicles and to figure out the factors that can influence consumers such as the cost of the vehicle, maintenance costs, charging infrastructure, government policy and demographic variables. Choice models, in general, have investigated whether consumers are willing to buy electric vehicles or not with given specific variables through the mixed logit model. Liao et al. (2017) mentioned that research using choice models could be biased due to the number of parameters, which makes a comparison for model fit more complex. The authors mentioned that there may be problems with self-selection bias, overfitting of variables and high correlations among variables. The authors also added that there is a limitation of demand forecasting in choice models in a way that makes it hard to convey psychological factors and uncertainty [1].

This paper conducts a survey on Suwon citizens to collect basic data to determine factors influencing consumer preferences for electric vehicles. Then, a specific structural equation model is established to identify variables that directly influence Suwon citizens' willingness to purchase an electric vehicle.

## 2. Literature Review

### 2.1. Research on Status of Electric Vehicle Market

Due to the escalated attention on environmental pollution, policy on emission standards has been fortified. This puts more pressure on the automobile industry, and it would be expected to focus on producing more eco-friendly vehicles [2]. As of 2020, there are 10 million electric vehicles in operation worldwide. The number of registered vehicles of all types decreased by 16% due to the COVID-19 pandemic, but the number of registered vehicles for electric vehicles increased by 41% compared to 2019 [3]. In addition, compared to 2017, when this research was conducted, 1 million electric vehicles were operated worldwide in 2017, and the number of registered electric vehicles increased by 54% compared to 2016 [4]. Of course, the increase in the number of registered units in 2017 was higher, but considering the total number of registered electric vehicles, the number of registered electric vehicles in 2020 is still high, which indicates that electric vehicle registrations are accelerating worldwide. Furthermore, despite the stagnancy in the automotive market due to COVID-19 in 2020, KAMA (Korean Automobile Manufacturers Association) announced that global sales of electric vehicles grew 46.1% in comparison to previous years. It is predicted that the world's electrical vehicle sales (CAGR) would increase by an average of 29% per year within the next 10 years between 2020 and 2030 [5]. As a result of the global automobile consumer survey, it was found that most consumers are considering purchasing an electric vehicle due to expectations of fuel cost reduction, concerns about climate change, and carbon emission reduction. It appeared that they were concerned about the inconvenience of use [6].

Until 2016, the electric vehicle market in South Korea was smaller than that of Canada, China, and European countries. However, in accordance with the government's electric vehicle purchase subsidy support policy in 2017, the number of registered vehicles in South Korea began to expand. In line with this, the government started to expand the electric vehicle charging infrastructure by installing 1139 fast chargers and 1387 slow chargers from 2017 [7]. It is expected that more than 10,000 electric vehicles will be distributed annually due to the government's market-motivating policy implementation [8]. As of 2018, the region where electric vehicles were introduced the most in South Korea was Jeju Island with 9167 units, followed by Gyeongsangbuk-do and Gyeongsangnam-do. In addition, Seoul, the metropolitan area, has 5919 units and Gyeonggi-do has 2263 units, indicating that the number of registered electric vehicles is continuing to increase [9]. Given this trend, it is expected that the number of registered electric vehicles in South Korea will continue to increase.

The biggest hurdles in the expansion of electric vehicles in South Korea are the expensive cost of the vehicle itself, limited driving range with a charged battery and long charging time [5,10]. Therefore, the government is actively subsidizing electric vehicles

for consumers to promote the distribution of EV nationally, such as special reductions in consumption tax and acquisition tax, discounts for electricity charging fees and discounts for expressway tolls.

### 2.2. Research on Factors Affecting Preference for Electric Vehicles Such as Characteristics of Electric Vehicle

Electric vehicles with their varying attributes often have a huge impact on consumer preferences. Many research results have shown that the higher the expense of purchasing an EV, the more negative the influence on preference for EVs [11–13]. On the other hand, operating cost also immensely influences propensity to buy an EV since the electricity for EVs is significantly cheaper than fossil-based fuel for internal combustion vehicles. Since operating cost includes both fuel efficiency and maintenance costs, a lower operating cost positively influences consumer preference [12]. Due to such lower operating cost, EVs are in an advantageous position compared to petroleum-based cars. Although purchasing an EV comes at a higher price than other vehicles, there is a competitive advantage in the long term due to energy efficiency. The total cost of ownership (TCO) is a fundamental concept and the relative TCO of EVs compared to that of other cars will directly influence the acceptance of EVs. However, despite these savings, many consumers are still inclined to avoid EV purchase [14]. Hence, consumers value current expenditure more than the long-term saving cost [15,16]. Besides the economic cost, there are other costs stemming from inconvenience, which can be intensified by either the driving range after one charge and/or the charging availability.

Electric cars use a battery that requires charging, so the availability and accessibility of the charging infrastructure influence the adoption of electric vehicles (EVs). These infrastructure attributes are found to have a positive impact on EV choice [1]. There are two methods for charging the battery: fast-charging and slow-charging. However, research so far has not differentiated the two methods and has not analyzed how these two methods influence consumers' preference for the purchase of electric vehicles.

Subsidy policy attributes are another influential factor in the purchase of an EV. Government subsidy on EV purchases can lower the purchasing price and enhance the t-value of using an EV. The purpose of the government policy to subsidize EV buyers is to promote EV automobile industries and to further prevent global warming by abiding by the $CO_2$ reduction agreement. There are slight differences in subsidy policy among countries with which policy is adopted for consumers. Especially in the United States and China, subsidies from both the central government and the local government policy have a considerable impact on EV purchase.

### 2.3. Research on Factors Affecting Preference for Electric Vehicles Such as Population Statistics

Many studies have been carried out on how differences in consumers' gender, age, education level, income level and occupation can influence the purchase of an electric vehicle. Generally, men show much more interest toward electric vehicles than women [17] and a higher intention to buy an EV [13,18–20]. One research study that focused on EV owners supports the finding that more men own EVs than women [21]. However, Liao et al. (2017) have pointed out that gender difference is not consistent for ownership of EVs [1]. Kim and Jeong (2018) also stated that gender has no influence on the purchase of an EV [22].

In terms of age difference, young people in general regard EVs as more attractive [17] and are more willing to buy them [11,20,23–25]. However, there are other studies which show middle-aged consumers have higher preference for EVs [19] and the age group between 20 and 60 have higher attraction towards EVs [14]. Given these points, younger consumers tend to be more innovative [17] and have higher awareness of environmental issues, which leads to them having higher preference for EVs [14,26].

Study results that include variables for the education level of consumers have been consistent. Higher education level is in accordance with EV attraction [27] and they show

higher intention to purchase an EV [11,13,21,23,25,28,29]. Research dealing with EV owners demonstrates that their education level is relatively high [29].

Income level as a variable toward EV preference shows controversial results. Although there is research that shows little correlation between income level and EV usage [11,25,30], there are other studies that show consumers with a higher education level are likely to be the early adopters of EVs [11,24,28,31]. One study that targeted early adopters of EVs showed that they are high income earners [29] and another study with Swedish EV owners showed their income levels are above the middle class [21]. Correspondingly, these findings support Morton's claim that higher income level is consistent with an innovative mindset [17].

There have not been consistent results regarding occupation type, but consumers with an occupation in the field of technology had higher interest in EVs [20,25]. Additionally, consumers who use EVs tend to have children [29] and one of the studies showed that the size of the family was between two and four [21]. The number of cars owned by the consumers showed inconsistent results [25] and Kim and Jeong (2018) claimed that ownership of car number overall has no significant influence on EV purchase [22]. However, consumers who already own a hybrid car have a higher tendency to purchase EVs [23].

## 3. Data

### 3.1. Review of Factors Affecting Electric Vehicle Purchase and Applied Variables

For deciding factors of purchasing EVs, Lieven et al. (2011) considered car price, driving range, car performance, durability, environmental impact and accessibility [32]. Zhang et al. (2011) applied purchasing timing and acceptability of price in addition to the above purchasing factors [33]. Junquera et al. (2016) analyzed the willingness to buy an electric vehicle in the Spanish market with major factors such as charging time, consumers' perception of the maintenance costs, driving range and driver's age [14]. Chu et al. (2017) analyzed their research on EV purchase factors with consumers' psychological tendency towards perception of the car, environmental concerns, innovativeness, uncertainty of driving range, and subjective knowledge level of EVs [34]. Degirmenci and Breitner (2017) have applied several variables such as environmental performance, price value, confidence range, attitude toward EV and purchase intention to apply to the structural equation model [35].

This research has reflected the fundamental factors influencing EV purchase from the structural equation model through the literature review and, further, included additional variables such as EV promotions, hill-start-assist control (HAC) system issues, consumers' perception of government subsidy and tax reductions for EV purchase and visiting systems for EV charging services. A review of the variables for the model is shown in Table 1.

**Table 1.** Review of variables for the model

| Author | Proposed Variables | Applied Variables |
|---|---|---|
| Lieven et al. (2011) | Price, Mileage, Performance, Durability, Convenience | |
| Zhang et al. (2011) | Purchase timing, Price acceptance | Age, Gender, Income, Incentives, Customer concerns, Issues |
| Janquera et al. (2016) | Life-cost cognition, Age, Mileage, Charging time | |
| Chu et al. (2017) | Psychological characteristics | |
| Degirmenci and Breitner (2017) | Environmental performance, Price value, Range confidence, Attitude toward electric vehicles, Purchase intention | |

### 3.2. Survey Overview and Basic Statistical Analysis

In this research, factors that are fundamentally related to EV purchase were derived through a literature review, and finally, factors affecting EV purchase were derived through expert Delphi research. Detailed EV-related matters such as publicity related to EVs, recognition of subsidy support and tax reduction, and visiting charging services were

additionally reviewed as variables and applied to the model. These additional suggested factors are considered to have a significant impact on the purchase of EVs by citizens.

The purpose of this survey was to investigate Suwon citizens' overall perceptions and concerns about EVs. First, the questionnaire was supplemented by conducting a preliminary survey on the contents of the pre-designed questionnaire for 10 people. Sufficient training was provided to the surveyor so that the survey was convenient with the supplementary questionnaire. The survey was conducted on 723 Suwon citizens and 719 samples, excluding 4 outliers, were used as analysis data. Additionally, the survey was conducted for two weeks in August 2017, and a face-to-face survey was conducted.

For the survey regarding perception of EVs and personal experience, a 5-point Likert scale was applied, and the survey results are shown in Table 2.

**Table 2.** Attributes of the respondents.

| Variable | | Number of Samples | % | Variable | | Number of Samples | % |
|---|---|---|---|---|---|---|---|
| Sex | Male | 450 | 62.5 | Members | 1 | 38 | 5.3 |
| | Female | 259 | 37.5 | | 2 | 76 | 10.6 |
| Age | 20s | 75 | 10.4 | | 3 | 292 | 40.6 |
| | 30s | 302 | 42.0 | | 4 | 265 | 36.9 |
| | 40s | 239 | 33.2 | | Over 5 | 46 | 6.4 |
| | 50s | 93 | 12.9 | Housing type | Detached house | 61 | 8.5 |
| | 60s | 10 | 1.4 | | Apartment | 492 | 68.4 |
| Driving experience | Under 5 years | 162 | 22.5 | | Row house | 97 | 13.5 |
| | 6 to 10 years | 195 | 27.1 | | Townhouse | 63 | 8.8 |
| | 11 to 15 years | 175 | 24.3 | | Etc. | 6 | 0.8 |
| | 16 to 20 years | 100 | 13.9 | Monthly household income | Less than USD 1000 | 3 | 0.4 |
| | Over 21 years | 87 | 12.1 | | USD 1000 to 2000 | 32 | 4.5 |
| Home ownership | Self | 419 | 58.3 | | USD 2000 to 3000 | 146 | 20.3 |
| | Rent: Charter | 248 | 34.5 | | USD 3000 to 4000 | 188 | 26.1 |
| | Rent: Monthly | 43 | 6.0 | | USD 4000 to 5000 | 180 | 25.0 |
| | Etc. | 9 | 1.3 | | Over USD 5000 | 170 | 23.6 |

### 3.3. Analysis of Survey Results

The survey results showed that the consumers' recognition level of EVs and government subsidy was about 37%. On the contrary, it was observed that about 32.3% of participants showed "poor understanding" about EVs. This means that the recognition level of EVs was polarized among the participants. The overall recognition level of EVs was, on average, 3.03 out of 5. This implied that citizens' awareness of EVs was standard.

Meanwhile, there were only 9.7% participants who had experience with EVs. One of the major opportunities for EV experience was through car sharing services. In contrast, participants who responded with a lack of experience with EVs stated this was due to a lack of opportunity to test drive EVs and/or lack of publicity channels to receive information about EVs. However, only 25.6% of participants showed a lack of experience with EVs, with no interest at all towards them. The levels of recognition of and experience with electric vehicles are shown in Table 3.

Participants' major concerns were battery-related issues, with an average of 4.01 out of 5. There was less concern with vehicle performance, but maintenance and repair cost concerns were relatively higher. Concerns regarding charging, excessive charging time and its inconvenience demonstrated the highest scores. In the related policy category, lack of publicity or opportunity to board scored the highest followed by lack of information on electric vehicles and infrastructure. Major concerns about electric vehicles are shown in Table 4.

**Table 3.** Level of recognition of and experience with electric vehicles.

| Variable | | Number of Samples | % | Variable | | Number of Samples | % |
|---|---|---|---|---|---|---|---|
| Overall recognition level of electric vehicle | Very well | 21 | 2.9 | Recognition level for purchasing subsidy | Very well | 19 | 2.6 |
| | Good | 245 | 34.1 | | Good | 193 | 26.8 |
| | Normal | 221 | 30.7 | | Normal | 237 | 33.0 |
| | Poor | 202 | 28.1 | | Poor | 223 | 31.0 |
| | Very Poor | 30 | 4.2 | | Very poor | 47 | 6.5 |
| Experience with electric vehicle | Yes | 649 | 90.3 | Inexperienced reason | Lack of test drive | 313 | 46.2 |
| | No | 70 | 9.7 | | Lack of publicity | 170 | 26.2 |
| | | | | | Not interested | 166 | 25.6 |

**Table 4.** Major concerns about electric vehicles.

| Variable | | Score | Variable | | Score |
|---|---|---|---|---|---|
| Battery | Mileage | 4.13 | Charging | Charging time | 4.11 |
| | Safety accident | 3.83 | | Charging cost | 3.6 |
| | Performance | 4.06 | | Charging-infrastructure breakdown | 3.88 |
| | Average | 4.01 | | Charging procedure | 4.1 |
| Related policy | Lack of information for electric vehicle and infrastructure | 3.81 | | Lack of charging infrastructure | 3.95 |
| | Lack of publicity or opportunity to board | 3.83 | | Average | 3.93 |
| | Complicated administrational procedures | 3.73 | Vehicle performance | Heating/Air-conditioning | 3.60 |
| | Model discontinuation or early termination of policy | 3.66 | | Hill start | 3.59 |
| | Average | 3.76 | | Lack of maintenance resources | 3.95 |
| | | | | Average | 3.71 |

*3.4. Implications of Survey Results*

　　Suwon citizens' perception of EVs is divided into two groups: one with high perception and another with low perception. Suwon city needs to continue to advocate for electric vehicles and incentives to encourage the low perception group to increase their perception further. Only 9.3% of citizens in Suwon city have had the opportunity to board an EV, which reveals the lack of accessibility to EVs. Among the participants with no EV boarding experience, only 25% of the participants gave the reason of lack of interest. Others pointed out the lack of opportunity to try to board an EV and/or the lack of publicity from companies and the government.

　　In terms of the experience path observed from participants with EVs, carsharing was the most popular approach followed by test driving events or by visit-and-experience opportunities through car dealerships. These findings suggest that there needs to be more active events and policies to provide EV experiences to citizens to enhance their perception of EVs.

　　Since most concerns about EVs were related to charging such as charging time, charging procedure and lack of charging infrastructure, this should be considered most importantly when promoting EVs.

**4. Model**

*4.1. Exploratory Factor Analysis*

　　The factors used in this study were based on the analytical study of EV purchase factors of 20 variables extracted from five research articles [14,32–35].

　　As a result of the factor analysis, they were classified into six categories, and the structural equation model was established with those six categories. Variables such as

lack of information, EV promotion policy, charging infrastructure, battery performance and safety, EV driving conditions, information on EVs, government subsidy policy, and visiting services were used to find out how each factor has an impact on the purchase of EVs. The hypotheses about this type of effect were set up and the test was performed using the structural equation model. In this paper, Cronbach's $\alpha$ was used to measure the reliability of the variables. It is generally believed that the internal consistency is higher when it reaches closer to 1, and if it exceeds 0.6, internal consistency is sufficient. Cronbach's $\alpha$ for each variable was calculated and resulted in: information on EV and vehicle policy (0.841), charging infrastructure (0.770), battery performance and safety (0.798), operating conditions (0.706), electric vehicle recognition (0.850), subsidy effect and charging service (0.763). Cronbach's $\alpha$ values for all the six selected variables were greater than 0.6, demonstrating that the reliability of the variables was acceptable. The factor analysis results are shown in Table 5.

**Table 5.** Factor analysis results.

| Factor | Component | | | | | |
|---|---|---|---|---|---|---|
| | 1 | 2 | 3 | 4 | 5 | 6 |
| Electric vehicle maintenance, repair, manpower shortage | 0.219 | 0.017 | 0.824 | 0.139 | 0.036 | 0.007 |
| Electric vehicle battery performance, battery price | 0.061 | 0.259 | 0.782 | 0.064 | −0.037 | 0.202 |
| Battery safety | 0.158 | 0.104 | 0.798 | 0.193 | 0.087 | −0.006 |
| Distance per charge | 0.017 | 0.755 | 0.106 | 0.161 | −0.022 | −0.015 |
| Driving on a ramp | 0.219 | 0.295 | 0.260 | 0.667 | 0.039 | −0.130 |
| Use of heaters and air conditioners | 0.233 | 0.275 | 0.199 | 0.727 | −0.046 | −0.059 |
| Insufficient charging infrastructure | 0.109 | 0.662 | 0.173 | 0.321 | −0.057 | 0.045 |
| Charging inconvenience | 0.161 | 0.776 | 0.047 | 0.096 | −0.058 | 0.047 |
| Lack of response information in case of failure | 0.130 | 0.009 | 0.029 | 0.520 | 0.063 | 0.082 |
| Charging price | 0.564 | −0.004 | 0.085 | 0.445 | 0.009 | −0.025 |
| Electrical cut-off/early termination of policy | 0.665 | 0.123 | 0.114 | 0.368 | 0.035 | 0.025 |
| Hassle-prone administrative procedures | 0.735 | 0.162 | 0.095 | 0.253 | −0.041 | 0.062 |
| Excessive charging time | 0.416 | 0.703 | 0.080 | −0.121 | −0.012 | 0.091 |
| Lack of detailed information on vendor information | 0.807 | 0.149 | 0.176 | 0.097 | 0.012 | −0.146 |
| Information shortage for electric vehicles and chargers | 0.796 | 0.153 | 0.206 | 0.071 | 0.006 | −0.141 |
| Vehicle price purchase intention | −0.376 | −0.158 | 0.132 | 0.009 | −0.362 | 0.124 |
| Impact of subsidy policy with purchase | −0.105 | 0.076 | 0.084 | −0.009 | 0.025 | 0.877 |
| Impact of visiting service for charging | −0.062 | 0.029 | 0.056 | 0.008 | −0.084 | 0.874 |
| Electric vehicle recognition | −0.050 | −0.071 | −0.007 | 0.059 | 0.894 | −0.051 |
| Benefits when purchasing | 0.022 | −0.110 | 0.141 | 0.021 | 0.884 | −0.019 |
| Cronbach's $\alpha$ | 0.841 | 0.770 | 0.798 | 0.706 | 0.850 | 0.763 |

As a result of the factor analysis, 20 factors were further categorized into 6 bigger categories. The first category of electrical vehicle information and policy is composed of 6 sub-factors: charging price, discontinuation of EV model and early termination of policy, hassle-prone administrative procedures, lack of detailed information on vendor information, information shortage for electric vehicles and chargers. The second category of battery infrastructure is composed of 4 sub-factors: distance per charge, insufficient charging infrastructure, charging inconvenience, excessive charging time. The third category of battery performance and safety is composed of 3 sub-factors: electric vehicle maintenance, repair, technician shortage, electric vehicle battery performance and battery price, and battery safety. The fourth category of EV driving environment is composed of 3 sub-factors: driving on a ramp, use of heaters and air conditioners, lack of response information in case of failure. The fifth category of perception on EV is composed of 2 sub-factors: electric vehicle recognition, and benefits when purchasing. The sixth category of battery service and government subsidy is composed of 2 sub-factors: impact of subsidy policy with purchase, and impact of visiting service for charging. Thus, 6 categories (EV information

and policy, battery infrastructure, battery performance and safety, EV driving environment, perception on EV, battery service and government subsidy) were used as latent variables and were applied to the structural equation model.

### 4.2. Research Hypothesis Setting

This research aims (1) to obtain all the factors that influence electrical vehicle purchase, (2) to analyze how much each latent variable affects the EV purchase, and (3) to identify the strongest factor that influences the decision to purchase an electric vehicle. Although Kim and Jeong (2018) suggested that socioeconomic variables such as number of vehicles owned, gender and age are correlated with EV purchase variables [22], other previous studies, such as Degirmenci and Breitner (2017), showed that most of the socioeconomic variables are not significant with the purchase of EVs [35]. Hence, those are not applied to this research model. The research hypotheses are shown in Table 6.

**Table 6.** Hypothesis verification of research hypotheses.

| | **Hypothetical Scheme** |
|---|---|
| Hypothesis #1 | *Information and government policy on EVs have effects on the decision of vehicle purchase.* |
| Hypothesis #2 | *Charging infrastructure has an effect on the decision of vehicle purchase.* |
| Hypothesis #3 | *Battery technology and safety have an effect on the decision of vehicle purchase.* |
| Hypothesis #4 | *Driving on ramps and internal systems such as heaters and AC have an effect on the decision of vehicle purchase.* |
| Hypothesis #5 | *Perceptions of both EV and tax reduction have an effect on the decision of vehicle purchase.* |
| Hypothesis #6 | *Tax reduction on the vehicle purchase and charging customer service have an effect on the decision of vehicle purchase.* |

### 4.3. Measurement Model Analysis

The Maximum Likelihood Method was used to estimate the parameters of the models. In order to evaluate the best fit, GFI (Goodness of Fit Index), AGFI (Adjusted Goodness of Fit Index), NFI (Normed Fit Index), IFI (Incremental Fit Index), TLI (Tucker–Lewis Index), CFI (Comparative Fit Index), RMSEA (Root Mean Square Error of Approximation), and RMR (Root Mean Square Residual) were used. Acceptable values for GFI, AGFI, NFI, IFI, TLI, and CFI > 0.90; RMR < 0.5; and RMSEA < 0.08. However, RMSEA < 0.05 can also be a good fit. The results of the earlier model showed that all indices were not satisfied.

In the measurement model, all the factor loadings ($\beta$), which compose the measuring tools, must have a minimum value of 0.5 or larger. As an outcome of early measurement model analysis, "EV information and policy" along with "charging price" did not meet the standards, so they were removed from the variables. Additionally, "lack of response information in case of failure" was removed from the variable since its factor loading ($\beta$) < 0.5. The modified measurement model is shown below in Figure 1.

### 4.4. Final Structural Equation Model

The research question posed in this research is the relationship between the latent variables (information shortage and EV supply policy, charging infrastructure, battery performance and safety, operating conditions, electric vehicle recognition, subsidy effect and charging service) and the decision to purchase an EV. The structural equation model was established for this relationship, and the results of the goodness of fit analysis are shown. The final model is shown below in Figure 2.

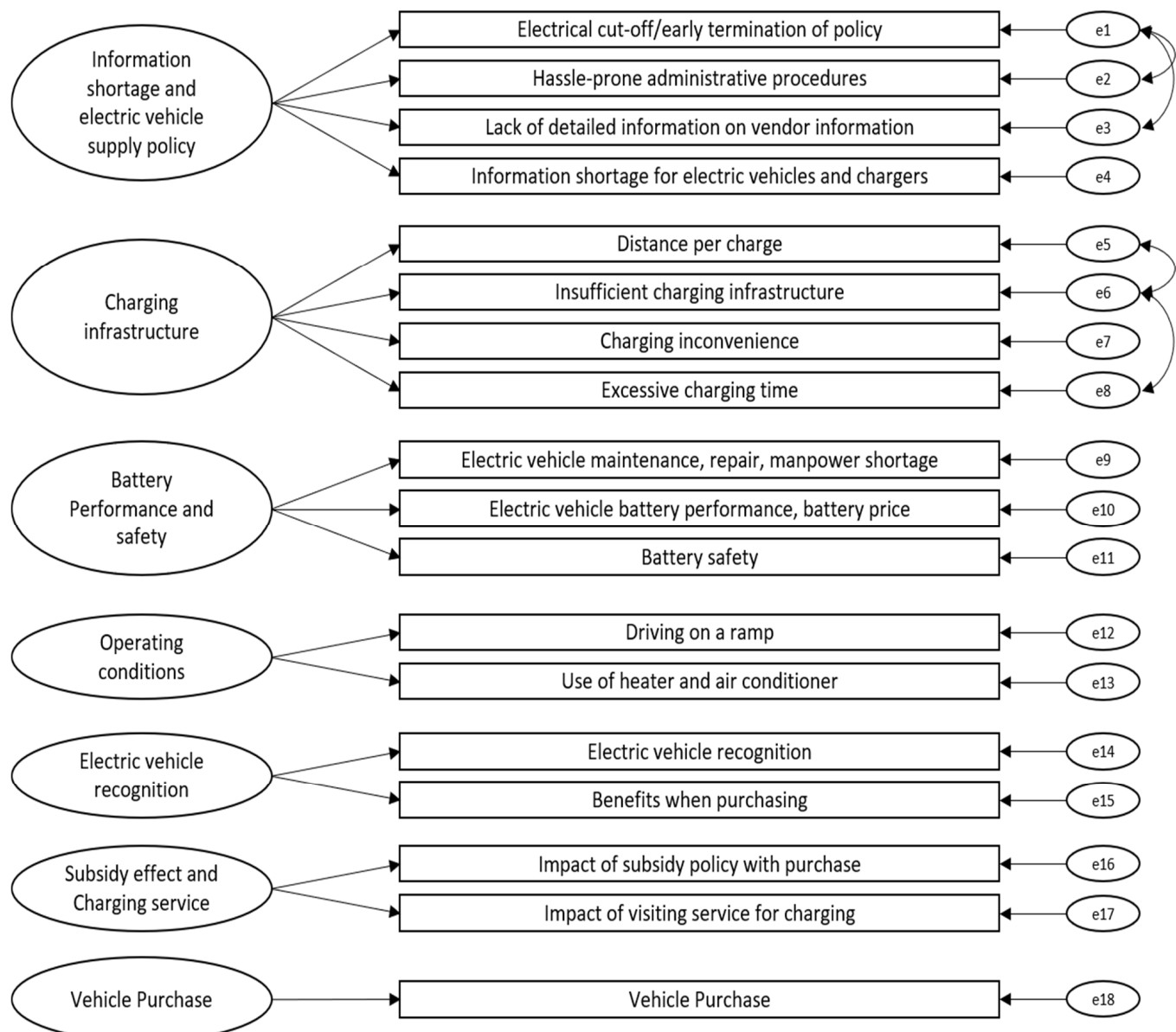

**Figure 1.** Modified measurement model.

### 4.5. Model Fit Analysis

Through the model fit analysis, the result of the goodness of fit index of CMIN/DF (=2.935) and RMSEA (=0.062) resulted lower than the reference, proving to be a good fit. The index of GFI (=0.941) and AGFI (=0.909) resulted higher than the reference. Increment fit index of IFI (=0.940), TLI (=0.916), CFI (=0.939), and NFI (=0.919) resulted higher than 0.9, so the final model's fit was found to be appropriate. The model fit analysis results are shown in Table 7.

### 4.6. Path Coefficient Analysis

For the results of the analysis, all values of CR (Construct Reliability) of the concept were above 0.70, demonstrating high reliability. Moreover, the validity of the model was verified through the values of AVE (Average Variance Extracted) of each concept, which scored above 0.50.

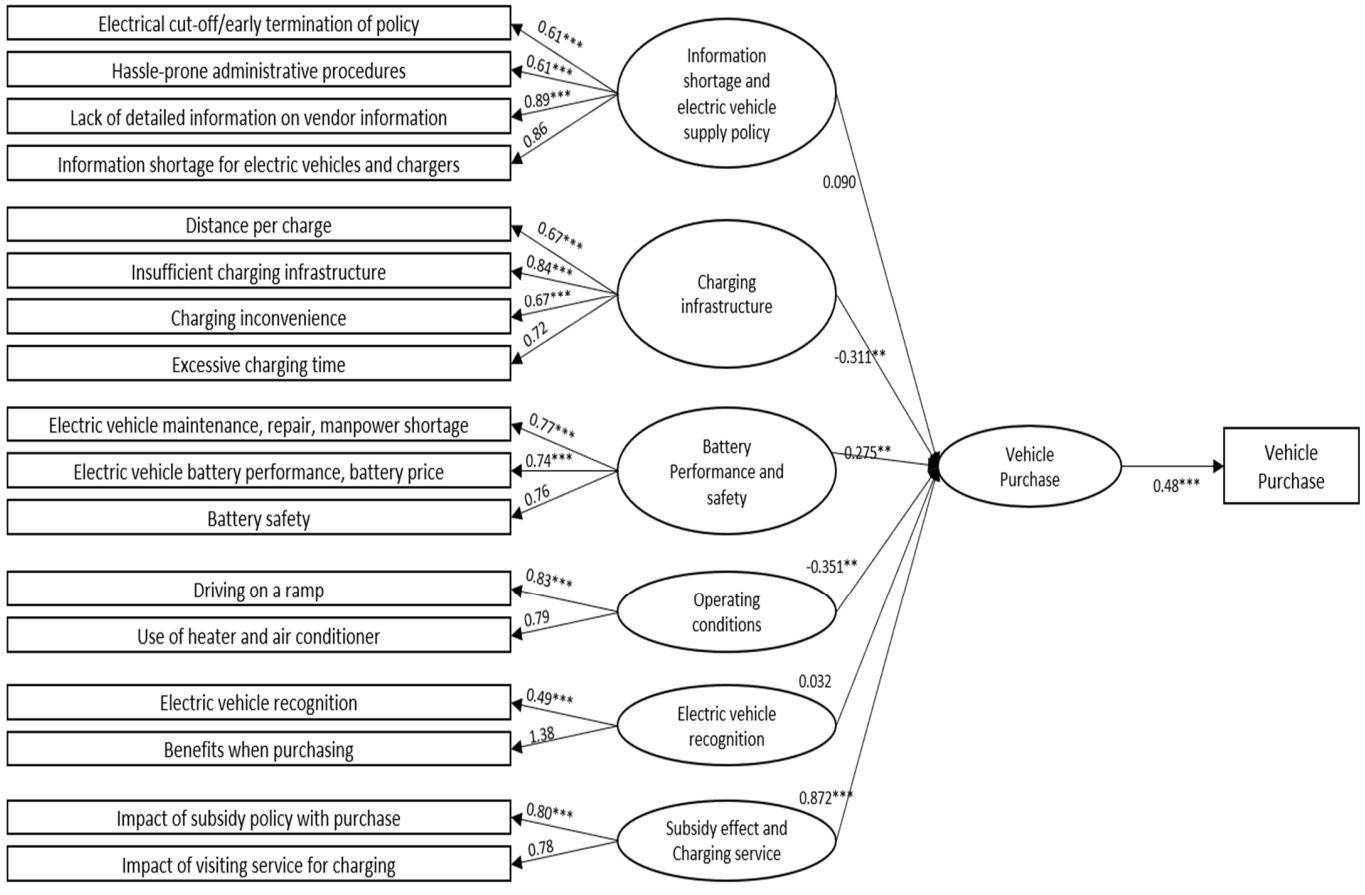

**Figure 2.** Final design model (standardized coefficient) (*** *p* < 0.01, ** *p* < 0.05).

**Table 7.** Model fit analysis results.

| Class | Absolute Fit Index | | | | | Increment Fit Index | | | |
|---|---|---|---|---|---|---|---|---|---|
| | CMIN/DF | GFI | AGFI | RMSEA | RMR | IFI | TLI | CFI | NFI |
| Final model | 2.935 | 0.941 | 0.909 | 0.062 | 0.036 | 0.940 | 0.916 | 0.939 | 0.919 |
| Thresholds Criterion | <3.00 Satisfaction | >0.9 Satisfaction | >0.8 Satisfaction | <0.1 Satisfaction | <0.05 Satisfaction | >0.9 Satisfaction | >0.9 Satisfaction | >0.9 Satisfaction | >0.8 Satisfaction |

The first category (Information shortage and electric vehicle supply policy) sub-factors had the most influence in the following order: lack of detailed information on vendor information, information shortage for electric vehicles and chargers, hassle-prone administrative procedures, and discontinuation of EV model and early termination of policy. The second category (Charging infrastructure) sub-factors had the most influence in the following order: insufficient charging infrastructure, excessive charging time, distance per charge followed by charging inconvenience. The third category (Battery performance and safety) sub-factors had the most influence in the following order: electric vehicle maintenance, repair, technician shortage, battery safety, electric vehicle battery performance followed by battery price. The fourth category (Operation conditions) sub-factors had the most influence in the following order: driving on a ramp followed by use of heaters and air conditioners. The fifth category (Electrical vehicle recognition) sub-factors had the most influence in the following order: benefits when purchasing, followed by electric vehicle recognition. The sixth category (Subsidy effect and charging service) had the most influence in the following order: impact of subsidy policy with purchase, followed by impact of visiting service for charging.

The first category (Information shortage and electric vehicle supply policy) showed β = 0.090 ($p > 0.005$) and a positive value (+), so it was not significant. The second category (Battery infrastructure) showed a negative correlation with β = −0.311 ($p < 0.005$) and a negative value (−), indicating that the more concerned the consumer is about battery infrastructure, the less likely they are to purchase an EV. The third category (Battery performance and safety) showed a positive correlation with β = 0.275 ($p < 0.005$), indicating that a higher recognition of battery performance and safety leads to an increase in EV purchase. Battery performance is related to one-time charging distance, which directly affects EV drivers. The fourth category (EV driving environment) showed a negative correlation with β = −0.3510 ($p < 0.005$). This indicates that the more concerned the consumer is about battery infrastructure, the less likely they are to purchase an EV. Electric vehicles tend to demonstrate effective fuel efficiency during the spring and fall seasons; however, ineffective fuel efficiency during summer and winter due to the use of air conditioners or heaters causes concern to drivers. Moreover, the capacity of the battery itself becomes low during summer and winter, since battery performance is very sensitive to temperature. The fifth category (Perception on EV) showed a positive correlation with β = 0.032 ($p > 0.005$) and had a positive value (+), but it had no significant value. The sixth category (Battery service and government subsidy) showed a positive correlation with β = 0.872 ($p < 0.001$) and a positive value (+). This indicates that higher government subsidy and the convenience of the battery service provided would result in more EV purchases. Thus, in order to expand the number of electric vehicles being distributed, there needs to be continuous subsidy support from the government as its policy. In addition, if a visiting charging service is provided when the EV is out of charge due to a lack of battery infrastructure, EV purchase rates would increase. The parameter estimation results for the hypotheses are shown in Table 8.

*4.7. Direct and Indirect Effect Analysis*

This study analyzed each latent variable's effect on EV purchase. Assessment of both direct and indirect effects revealed that for EV purchase, government subsidy and visiting charging services have the largest impact followed by EV driving environment, charging infrastructure, battery performance and safety, and information shortage and electric vehicle supply policy. However, electric vehicle recognition and subsidy recognition effects demonstrated the least impact on EV purchase. Direct and indirect effect analysis results are shown in Table 9.

*4.8. Research Hypotheses Verification Results*

This research established hypotheses and presents their acceptance or rejection (no acceptance). The first and fifth hypotheses of this research had no influence on EV purchase and thus are rejected. However, the rest of the hypotheses (2, 3, 4, and 6) are accepted, since it was shown that they affect EV purchase. Hypotheses verification is shown in Table 10.

**Table 8.** Path coefficient analysis.

| | | | Non-Standardization Coefficient (B) | Standardization Coefficient (β) | S.E. | t | p | CR | AVE |
|---|---|---|---|---|---|---|---|---|---|
| Information shortage and electric vehicle supply policy | → | Electrical cut-off/early termination of policy | 0.805 | 0.611 | 0.051 | 15.744 | *** | | |
| | → | Hassle-prone administrative procedures | 0.751 | 0.614 | 0.044 | 17.24 | *** | 0.879 | 0.652 |
| | → | Lack of detailed information on vendor information | 1.057 | 0.89 | 0.042 | 25.155 | *** | | |
| | → | Information shortage for electric vehicles and chargers | 1 | 0.858 | | | | | |
| Charging infrastructure | → | Distance per charge | 0.917 | 0.668 | 0.067 | 13.721 | *** | | |
| | → | Insufficient charging infrastructure | 1.238 | 0.837 | 0.084 | 14.767 | *** | 0.868 | 0.623 |
| | → | Charging inconvenience | 0.913 | 0.665 | 0.064 | 14.328 | *** | | |
| | → | Excessive charging time | 1 | 0.719 | | | | | |
| Battery performance and safety | → | Electric vehicle maintenance, repair, manpower shortage | 0.912 | 0.772 | 0.051 | 17.41 | *** | | |
| | → | Electric vehicle battery performance, battery price | 0.895 | 0.735 | 0.051 | 17.925 | *** | 0.869 | 0.689 |
| | → | Battery safety | 1 | 0.76 | | | | | |
| Operating conditions | → | Driving on a ramp | 1.082 | 0.825 | 0.06 | 17.905 | *** | 0.86 | 0.754 |
| | → | Use of heaters and air conditioners | 1 | 0.794 | | | | | |
| Electric vehicle recognition | → | Electric vehicle recognition | 0.353 | 0.49 | 0.147 | 2.404 | 0.016 ** | 1.04 | 1.068 |
| | → | Benefits when purchasing | 1 | 1.375 | | | | | |
| Subsidy effect and charging service | → | Impact of subsidy policy with purchase | 1.11 | 0.797 | 0.09 | 12.342 | *** | 0.77 | 0.625 |
| | → | Impact of visiting service for charging | 1 | 0.776 | | | | | |
| Information shortage and electric vehicle supply policy | → | | 0.029 | 0.09 | 0.035 | 0.848 | 0.397 | | |
| Charging infrastructure | → | | −0.116 | −0.311 | 0.037 | −3.097 | 0.002 ** | | |
| Battery performance and safety | → | Vehicle purchase | 0.097 | 0.275 | 0.038 | 2.522 | 0.012 ** | | |
| Operating conditions | → | | −0.126 | −0.351 | 0.046 | −2.747 | 0.006 ** | | |
| Electric vehicle recognition | → | | 0.005 | 0.032 | 0.008 | 0.641 | 0.522 | | |
| Subsidy effect and charging service | → | | −0.26 | 0.872 | 0.029 | 8.834 | *** | | |

*** $p < 0.01$, ** $p < 0.05$

**Table 9.** Direct and indirect effect analysis results.

| Latent Variable | | Information Shortage and Electric Vehicle Supply Policy | Charging Infrastructure | Battery Performance and Safety | Operating Conditions | Electric Vehicle Recognition | Subsidy Effect and Charging Service |
|---|---|---|---|---|---|---|---|
| Vehicle purchase | Direct effect | 0.090 | −0.311 | 0.275 | −0.351 | 0.032 | 0.872 |
| | Indirect effect | 0.043 | −0.149 | 0.132 | −0.169 | 0.016 | 0.419 |

**Table 10.** Hypotheses verification.

| | **Hypothetical Scheme** | |
|---|---|---|
| Hypothesis #1 | *Information and government policy on EVs have effects on the decision of vehicle purchase.* | Rejected |
| Hypothesis #2 | *Charging infrastructure has an effect on the decision of vehicle purchase.* | Accepted |
| Hypothesis #3 | *Battery technology and safety have an effect on the decision of vehicle purchase.* | Accepted |
| Hypothesis #4 | *Driving on ramps and internal systems such as heaters and AC have an effect on the decision of vehicle purchase.* | Accepted |
| Hypothesis #5 | *Perceptions of both EV and tax reduction have an effect on the decision of vehicle purchase.* | Rejected |
| Hypothesis #6 | *Tax reduction on the vehicle purchase and charging customer service have an effect on the decision of vehicle purchase.* | Accepted |

## 5. Discussion

Higher government subsidy costs and more support from the government in terms of charging service directly and positively influence the decision to purchase an electrical vehicle and promote their expansion. Therefore, continuous support from the local government and Department of Environment is needed for the expansion of electrical vehicles. However, this kind of subsidy policy may not be sustainable. At present, in South Korea, the central government is offering subsidies on EV purchase, and local governments are also offering subsidies on EV purchases depending on financial conditions. Local governments are differentiating subsidy by type of vehicle: passenger car, cargo (compact, small, small special). By promoting these kinds of subsidy policies, central and local governments are also aiming to implement a 2050 Carbon Neutral Strategy. However, unlike these implementation efforts, the budget for EV supply is insufficient each year to meet the demand for EV purchases. To improve the sustainability of EV purchase subsidy policies, more specific considerations may be needed. Moreover, as a solution to the lack of charging infrastructure, offering charging services through direct visits in cases of emergency would increase vehicle purchase even more. Kim and Jeong (2018) claimed that the most influential variables on purchase are real vehicle price, subsidy policy and visiting service for charging. Among the variables, visiting service for charging and subsidy policy have the biggest impact on the purchase [22].

As shown from the results, support from the government directly influences EV purchase. Government subsidy on the EV purchase can lower the purchasing price and enhance the economic value of using an EV. The purpose of the government policy to subsidize EV buyers is to support the promotion of the automobile industries and to further prevent global warming by abiding by the CO2 reduction agreement. There are slight differences in subsidy policy among countries with which policy is adopted for consumers. Especially in the United States and China, subsidies from both central government and local government policies have a considerable impact on EV purchase. Liao et al. (2017) reported that reductions in tax at the time of EV purchase are remarkably effective for consumers' preference for EV; other benefits that come with an EV purchase are not as effective as tax reduction [1].

Moreover, electric cars use a battery that requires charging, so the availability and accessibility of the charging infrastructure influence the adoption of electric vehicles (EVs). These infrastructure attributes are found to have a positive impact on EV choice (Liao et al., 2017) [1]. The results from our research also indicate that the more concerned the consumer is about battery infrastructure, the less likely they are to purchase an EV. Therefore, more charging infrastructure needs to be set up for the EV consumers. There are two methods for charging the battery: fast-charging and slow-charging. However, research so far has not differentiated how these two methods influence consumers' preferences for the purchase of electric vehicles.

Charging infrastructure needs to be set up according to the precise data collected from the distribution of electric cars. Public charging infrastructure needs to focus on the location where the demand is the highest from the beginning. Additionally, it needs to be set up in an accessible way for emergency charging and road charging as well.

Locations where a public fast-charging station would be acceptable include public parking lots, gas stations, public institutions, big malls, major traffic areas and facilities that attract crowds. In contrast, public slow-charging stations should be built nearby apartment complexes, industrial complexes and facilities that attract an average amount of people.

The results from this study suggest that higher recognition of battery performance and safety leads to an increase in EV purchase. Battery performance is related to one-time charging distance, which directly affects the EV driver. Degirmenci and Breitner (2017) indicated that EVs' high price and the short one-time changing distance are the two major factors that prevent the expansion of the EV usage [35].

Moreover, the results indicated that the higher the concern of the EV driver with driving on a ramp and operation of internal systems such as heaters and AC, the larger the negative effect on the decision to purchase a vehicle. Electric vehicles tend to demonstrate effective fuel efficiency during the spring and fall seasons; however, ineffective fuel efficiency during summer and winter due to the use of an air conditioner or heater can possibly cause concern for drivers. Moreover, the capacity of the battery itself becomes low during summer and winter, since battery performance is sensitive to temperature.

Policies that are related to information about EVs do not influence EV purchase. This indicates that advertisements may not be directly related to the purchase of EVs. Additionally, perception of EVs and perception of subsidy on EV purchase do not influence EV purchase. This indicates that provision of a subsidy on EV purchase has more influence compared to the perception of subsidy on EV purchase.

## 6. Conclusions

This paper has investigated the factors that influence electric vehicle purchase through surveys from citizens of Suwon city. With the analysis of survey data, hypotheses were established, and the effects of the factors on EV purchase were clarified. For model fit analysis, the results of the goodness of fit index of CMIN/DF (2.935) and RMSEA (0.062) were lower than the reference, proving to be a good fit, and the index of GFI (0.941) and AGFI (0.909) were higher than the average. The increment fit index of IFI (0.940), TLI (0.916), CFI (0.939), and NFI (0.919) were higher than 0.9, so the fitness of the final model was found to be satisfied.

The information shortage and electric vehicle supply policy (first category) showed $\beta = 0.090$ ($p > 0.005$) and a positive value (+), so it was not significant. Battery infrastructure (second category) showed a negative correlation, meaning they were less likely to purchase an EV due to this. Battery performance and safety (third category) showed a positive correlation with $\beta = 0.275$ ($p < 0.005$), indicating that a higher recognition of battery performance and safety leads to an increase in EV purchase. Thus, battery performance is related to one-time charging distance, which directly affects EV drivers. EV driving environment (fourth category) showed a negative correlation with $\beta = -0.3510$ ($p < 0.005$). This indicates that the more concerned the consumer is about battery infrastructure, the less likely they are to purchase an EV. Electric vehicles tend to demonstrate effective

fuel efficiency during the spring and fall seasons; however, ineffective fuel efficiency during summer and winter due to the use of air conditioners or heaters causes concern for drivers. Moreover, the capacity of the battery itself becomes low during summer and winter, since battery performance is very sensitive to temperature. Perception of EVs (fifth category) showed a positive correlation with β = 0.032 ($p > 0.005$) and was not significant. Battery service and government subsidy (sixth category) showed a positive correlation with β = 0.872 ($p < 0.001$). This indicates that higher government subsidy and the convenience of the battery service provided would result in more EV purchases. Thus, to expand the number of electric vehicles being distributed, there needs to be continuous subsidy support from the government as its policy. In addition, if a visiting charging service is provided when the EV is out of charge due to a lack of battery infrastructure, EV purchases would increase.

For the analysis of each latent variable's effect on EV purchase, the assessment of both direct and indirect effects revealed that for EV purchase, the government subsidy and visiting charging service had the largest impact followed by EV driving environment, charging infrastructure, battery performance and safety, and information shortage and electric vehicle supply policy. However, electric vehicle recognition and subsidy recognition effects showed the least impact on EV purchase.

To fundamentally expand the supply of EVs, it seems that related technology needs to be improved. According to the research results, it was found that the performance and safety of EV batteries, especially where there are many slopes in the EV driving route, and the decrease in driving performance due to the operation of heaters and air conditioners during driving, influence the purchase of an EV. Due to the characteristics of a battery-powered EV, it can be seen that the performance of a battery-powered vehicle is somewhat inferior. Additionally, when heaters and air conditioners are operated, the battery performance may deteriorate. In order to supplement the limitations of EVs, EV technology needs to be improved from its current level. In addition, if these improvements are made, it is expected that the sales rate of EVs will increase.

The limitation of this research is that the research is focused on Suwon citizens. It would be inappropriate to generalize the results of this research to other cities in South Korea as well as other countries. Thus, additional research may be conducted similarly on more cities and countries. Additionally, it seems that it is necessary to systematically analyze consumer choice factors or decision making that affects EV purchase. As a result, findings from this research can be used further to make policies with the purpose of expanding the purchase of electric vehicles.

**Author Contributions:** Conceptualization, S.K., J.C. and Y.Y.; methodology, S.K. and Y.Y.; software, S.K.; validation, S.K.; formal analysis, S.K.; investigation S.K., J.C., Y.Y. and H.K.; resources, S.K. and H.K.; writing-original draft preparation, S.K., J.C. and H.K.; writing review and editing, S.K. and H.K.; visualization, S.K., J.C. and H.K.; supervision, S.K. and H.K.; project administration, S.K. All authors have read and agreed to the published version of the manuscript.

**Funding:** This research was funded by Suwon Research Institute in Korea, grant number SRI-policy study-2017-09.

**Institutional Review Board Statement:** Not applicable.

**Informed Consent Statement:** Informed consent was obtained from all subjects involved in the study.

**Data Availability Statement:** Not applicable.

**Conflicts of Interest:** The authors declare no conflict of interest.

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
