# Peer review of "Analysis of Influencing Factors in Purchasing Electric Vehicles Using a Structural Equation Model: Focused on Suwon City"

_sustainability, doi:10.3390/su14084744_

Round 1

Reviewer 1 Report

General remarks:

The authors did interesting research about connecting the EV market with the SEM model. On the one hand, the applied methodologies are clear and well adopted, the results are clearly described and discussed. On the other hand, what are the reasons for choosing the SEM for the investigation? The SEM is typically applied to phenomenons (latent variables) that cannot be directly measured. However, measuring willingness has broad scientific literature. In the SEM, the prediction of latent variables is in focus, not the measurable variables. The authors use the SEM to determine the variables that influence the citizens' willingness to purchase an electric vehicle. They group the measurable variables and call them latent variables. Please reason why SEM is the best choice to investigate the influencing factors? What are other potentially applicable methods to analyze the effects? How does SEM have advantages over them? Don't the authors think that comparing the SEM results with available statistics would be necessary?

The results are based on a city survey, which should be highlighted in the paper's title.

Chapter 2 and Chapter 5: a further intensive literature review is needed to have a general picture about the latest market development/status and the investigated influences (worldwide examples are dated to 2017). The authors' statements need more verification of recent international scientific results.

The survey methodology is soundly applied. Why did the authors neglect the price factor from the survey?

Specific remarks:

Chapter 3.1: Table 1. isn't cited in the text.

Line 198: Table 1. please shortly explain the "applied variables" column

Chapter 3.3: Table 3. and Table 4. aren't cited in the text.

Chapter 4.1: Table 5. isn't cited in the text.

Line 271: Table 5., please shorty explain the "component" columns (1-6)

Line 331: Figure 1., please shortly explain the "e" (error) parameters

Chapter 4.5: Table 6. isn't cited in the text.

Chapter 4.7: Table 8. isn't cited in the text.

Chapter 4.8: Table 9. isn't cited in the text.

Line 414-415: the authors stated, "continuous support from the local government and department of environment is needed for the expansion of electrical vehicle" – how can such policy be maintained in the long term? It seems to be unrealistic to continuously subsidize EV purchases because that leads to market distortions and market failures.

Line 425: please revise the text

Reviewer 2 Report

The paper An Analysis for Influencing Factors in Purchasing Electric Vehicle Using a Structural Equation Model deals with a real and important topic, which is consistent with the purpose of the journal.
However, it needs revisions before it can be considered for publication in the journal Sustainability. The following comments should be considered.
1. First of all, the citation does not agree with the journal template, two forms of citation are used at the same time.
2. Point 3 should be redefined - Research Methodology or Materials and Methods - which includes, the objectives and hypoteses, the research model
3. The period when the data were collected is quite old, the year 2017, the market of eco vehicles and the perception of consumers has changed quite a lot in 4 years
4. The discussion section could compare the results obtained with similar previous research and present the authors' opinion on the differences.
5. In conclusion, the theoretical and managerial implications of the study, as well as the limits of the research must be mentioned

Round 2

Reviewer 1 Report

Thank the Authors for the reply and the modifications. I accept them all.

Author Response

Thank you for the kind review.

Reviewer 2 Report

Dear authors, the efforts to improve the article are obvious. The conclusions could be filled up with more recommendations. I recommend publishing the article after minor revisions.

Author Response

Thank you for the kind review.
